# A Comparison of Barrier Factors between Hospitals with and without Cardiac Rehabilitation Programs in Korea: A Nation-Wide Survey Research

**DOI:** 10.3390/jcm11092560

**Published:** 2022-05-02

**Authors:** Chul Kim, Jidong Sung, Jae-Young Han, Sungju Jee, Jang Woo Lee, Jong Hwa Lee, Won-Seok Kim, Heui Je Bang, Sora Baek, Kyung-Lim Joa, Ae Ryoung Kim, So Young Lee, Jihee Kim, Chung Reen Kim, Oh Pum Kwon

**Affiliations:** 1Department of Rehabilitation Medicine, Inje University Sanggye Paik Hospital, Seoul 01757, Korea; josephck@naver.com; 2Division of Cardiology, Department of Medicine, Sungkyunkwan University School of Medicine, Seoul 06351, Korea; jidong.sung@gmail.com; 3Department of Physical Medicine and Rehabilitation, Chonnam National University Medical School and Hospital, Gwangju 61469, Korea; 4Department of Rehabilitation Medicine, Chungnam National University College of Medicine, Daejeon 35015, Korea; jeesungju@gmail.com; 5Department of Physical Medicine and Rehabilitation, National Health Insurance Service Ilsan Hospital, Goyang 10444, Korea; medipia@gmail.com; 6Department of Physical Medicine and Rehabilitation, Dong A University College of Medicine, Busan 49201, Korea; jhlee08@dau.ac.kr; 7Department of Rehabilitation Medicine, Seoul National University Bundang Hospital, Seongnam 13620, Korea; wondol77@gmail.com (W.-S.K.); pum78@naver.com (O.P.K.); 8Department of Rehabilitation Medicine, Chungbuk National University Hospital, Cheongju 28644, Korea; bang@chungbuk.ac.kr; 9Department of Rehabilitation Medicine, Kangwon National University School of Medicine, Chuncheon 24289, Korea; sora.baek@kangwon.ac.kr; 10Department of Rehabilitation Medicine, Inha University Hospital, Incheon 22332, Korea; drjoakl@gmail.com; 11Department of Rehabilitation Medicine, School of Medicine Kyungpook National University, Daegu 41944, Korea; ryoung20@hanmail.net; 12Department of Rehabilitation Medicine, Jeju National University School of Medicine, Jeju 63241, Korea; bluelsy900@hanmail.net; 13Department of Rehabilitation Medicine, Wonkwang University School of Medicine, Iksan 54538, Korea; gold82mouse@hanmail.net; 14Department of Rehabilitation Medicine, Ulsan University College of Medicine, Ulsan 44033, Korea; crkim@uuh.ulsan.kr

**Keywords:** cardiac rehabilitation, barriers, percutaneous coronary interventions, survey

## Abstract

The percentage of patients undergoing cardiac rehabilitation programs (CRP) is very low (30–40%), and hospitals providing CRP are either insufficient or lacking, even in countries with advanced medical care; therefore, this study aims to investigate the barriers, as well as compare the differences between hospitals, with or without CRP. We conducted a survey, in which the questionnaire was distributed through post or email to 607 specialists who work at 164 hospitals performing percutaneous coronary interventions (PCI). The results were as follows: (1) of the 164 hospitals, 132 responded (response rate: 80.5%). While all 47 hospitals with CRP responded (100%), from among the 117 hospitals without CRP, 85 responded (72.7%). (2) Of the 607 specialists, 227 responded (response rate: 36.9%). The response rates according to specialties were as follows: cardiologists (28.9%), cardiac surgeons (24.5%), and physiatrists (64.1%). (3) While the specialists at hospitals with CRP identified patient referral, transportation, and cost as the major barriers, for those at hospitals without CRP, all items were considered barriers, especially the items related to equipment, space, workforce, and budget as being more serious barriers. Therefore, in order to actively promote CRP, it is suggested that governments consider the customized support system according to the performance of CRPs.

## 1. Introduction

Cardiovascular diseases (CVD) are among the leading global causes of mortality and morbidity, with a rising incidence in low and middle-income countries [1]. In Korea, too, the CVD mortality rate is steadily increasing, and CVD is still the second largest cause of death, with 63 deaths per 100,000 people [2].

Cardiac rehabilitation (CR) is a program of exercise, education, and counseling designed to help you recover after CVD. Phase II of the cardiac rehabilitation program (CRP) lasts for up to 6–12 weeks, and the price varies greatly in each country [3].

CR—an essential treatment to improve physical functions, quality of life, and prognosis (including recurrence and mortality)—is currently implemented in 111 countries worldwide [4]. It has been shown to reduce all-cause and cardiac mortality by 13–26% and 20–36%, respectively, as well as decrease patients’ five-year all-cause mortality by 21–34% [5]. On account of these significant benefits, the AHA/ACC gives CR either a class IA or IB recommendation for secondary prevention in almost all cases of CVD, after vital signs stabilize [6]. Recently, as a way to remind physicians of CR and facilitate patient participation, clinical practice guidelines for CR in Korea have reported that it has already become a universal treatment [7]. Although CR has been reported to decrease five-year mortality by 59% in acute myocardial infarction (AMI) patients, who had participated in it, as compared to those who had not, Korea’s CR participation rate is still very low at 1.5%, and it is 20–30% even in countries with advanced medical care [8,9,10]. Every country, including Korea, is striving to improve its participation rates in CRP.

Although there are various reasons for the low participation rates, one of them is that presently there are only a few hospitals providing CRP. Therefore, it is necessary and important to investigate the current status of hospitals that do not provide CRP and identify the differences in barriers between the hospitals, with or without CRP.

## 2. Materials and Methods

### 2.1. Target Subjects and Conducting of the Survey

To determine the actual status of CRP, the survey questionnaire was distributed either by post or email to 607 specialists—308 interventional cardiologists, 143 cardiac surgeons, and 156 physiatrists—working in 164 hospitals that performed the percutaneous coronary intervention (PCI). Target hospitals were selected when insurance claims of PCI were registered with the Health Insurance Review and Assessment Service (HIRA), which is an independent agency responsible for the claims review and quality assessment of the National Health Insurance. Specialists have professional licenses recognized by the Minister of Health and Welfare in their field. In particular, interventional cardiologists must have at least two years of experience and be currently performing PCI. They included specialists who were not excluded, depending on whether they were actually playing a role in the cardiac rehabilitation team. If more than 60% of the questions were answered, it was counted as a respondent, and the hospital where the respondent works was also counted as the responder hospital. Hospitals sending in responses from one or more specialists were considered as having responded. All the specialists’ responses were used for conducting the comparative analysis, relating to the questions in the assessment checklist for identifying the barriers and investigating the status of CR implementation. However, responses with inaccurate reply codes or inapplicable answers were excluded from the analysis. Table 1 shows the survey questionnaire that was used to investigate the barriers to the activation of CRP. For statistical analysis, the responses of the 227 specialists working in 132 hospitals that performed the PCI were converted into scores, using a 5-point Likert scale, ranging from 1 (not at all) to 5 (very much). 

### 2.2. Statistical Analysis

The chi-square test and Student’s t-test were used to compare the proportions and values between the hospitals, with and without CRP, as well as to compare physiatrists with other specialists. All statistical analyses were performed using the Statistical Package for the Social Sciences (SPSS) program (version 27.0; IBM SPSS, Armonk, NY, USA). The significance level was set at a *p*-value lower than 0.05.

### 2.3. Ethics Statement

This study was approved by the Institutional Review Board of Chonnam National University Hospital (CNUH-2020-163) and other participating centers. The need to obtain informed consent was waived due to the study’s non-clinical and minimal-risk nature. Since participants’ personal information (name, address, ID, phone number, hospital ID) was not collected, their anonymity was preserved.

## 3. Results

### 3.1. The General Characteristic between the Hospitals with and without CRP

In hospitals with CRP, various procedures and surgeries were performed, and the annual number of PCI is high (Table 2).The period of cardiac rehabilitation in hospitals with CRP was the most between 3 and 6 years (Table 2).

### 3.2. The Response Rate to the Survey

Of the 164 hospitals, 132 responded (response rate: 80.5%). All 47 hospitals with CRP responded (100%), whereas of the 117 hospitals without CRP, 85 responded (72.6%) (Figure 1).Of the 607 specialists, 227 responded (response rate: 36.9%). The response rates according to specialty were as follows: cardiologists (28.9%), cardiac surgeons (24.5%), and physiatrists (64.1%) (Figure 2).

### 3.3. Barriers Based on the Results of the Survey

All items except for patient referral were the differences between hospitals with or without CRP. Moreover, the transportation (mode and distance) items, unlike other items, were a barrier that was considered more serious in hospitals with CRP. Whereas specialists in hospitals without CRP considered all the items as barriers, especially the items related to equipment, space, workforce, and budget, which were more serious barriers. (Table 3).

Differences between rehabilitation medicine specialists and other specialists in hospitals with CRP were seen in the following items: Lack of Equipment for CRP, Lack of Budget for CRP, Patient Transportation: Mode and Distance (Table 4).

Differences between rehabilitation medicine specialists and other specialists in hospitals without CRP were seen in the following items: Lack of Equipment for CRP and Lack of Space for CRP (Table 5).

## 4. Discussion

A study on the global status of CR density was recently published [4]. It revealed that although CR is strongly recommended for patients with acute coronary diseases, only half of the countries are performing it. Furthermore, as compared to the number of patients who need CR [11], it is being provided insufficiently. For Korea, too, CR capacity and density were provided. This report calculated CR capacity based on the median number of patients that each facility could serve per year (=250) multiplied by the number of facilities, and CR density based on the AMI incidence divided by the CR capacity. In Korea, after being discharged from hospitals, phase II CR was provided only to approximately 1.5% of patients with AMI (960 out of 64,982 patients), and the CR density was approximately 10. Kim et al. reported that about 260 CR institutions were needed in Korea, considering the rate of patients with AMI [9]. In light of the number of patients with heart failure, or other diseases that necessitate CR, more institutions will be needed. Furthermore, the insufficiency of CR institutions is a global issue, and not just in Korea. We, therefore, believe that it will be noteworthy to undertake a comparison of barriers between institutions that have CRP and those that do not.

Depending on the response request method, the response rate may be affected and there is a possibility that it may affect the result. However, since the survey was at the time of the outbreak of COVID-19, both post and electronic questionnaires were used to increase the response rate. In this study, the survey’s response rate of 36.9%, was lower than expected. This could have been due to insufficient time for participation, as the survey period coincided with an increase in the COVID-19 pandemic. Moreover, it may have also resulted from a lack of interest in CR. Regardless of this, specialists working at hospitals with CRP had a 15% higher response rate than those working at hospitals without CRP. This may be the result because the degree of interest and attentiveness in cardiac rehabilitation had an effect on the response rate to increase.

In a hospital without CRP, in fact, all factors are the important barriers, and this is also shown in our study results. However, the point to pay attention to here seems to be the reason for not being able to start the CRP, especially in the facility (equipment, space, workforce). In order to solve this problem, we suggest that it will be necessary for the national or the government to invest in an initial facility in order to allow many hospitals to open cardiac rehabilitation centers.

Another important finding from this study is that the specialists who worked at hospitals with CRP considered the transportation of patients as the biggest barrier. Therefore, this issue is likely to be considered one of the biggest problems when actively promoting CRP in the future. To address this, implementation of home-based CR, as reported by Chindlhy, or more active participation in community-based CR, is necessary. It will be important to recognize the need for help from the government and society to increase the number of hospitals that can provide CR in more regions [6]. This study’s survey results from hospitals without CRP show more serious problems. Most criteria received scores greater than 4.0, which suggests that all the items in the questionnaire were needed. In particular, the survey showed that space, staff, and equipment were significantly under-supported in hospitals without CRP, compared to those with CRP. From this finding, we can infer the reasons for CRP not being offered in hospitals. Notably, there has been a recent, similar report from an investigation of hospitals without CRP, but which is limited in that the number of participating hospitals was small and only cardiologists participated [12].

In the comparison between specialists, there were several items with statistical differences, but the difference in actual scores was not as large as 0.5, so it is not considered to have great clinical significance. Among the results, in hospitals with CRP, physiatrists answered that the equipment was low as a barrier compared to other specialists, and in hospitals without CRP, the importance of equipment was high.

A limitation of this study is that, while it was a national survey (including urban and local areas), it was restricted to Korea. However, its results can be applied to other developing countries and could increase awareness of the need for additional global studies. It is also necessary to conduct future research involving various team members, such as general practitioners, exercise therapists, nutritionists, etc. Another limitation is that the role of the specialist in a cardiac rehabilitation team cannot be presented separately. This is because cardiac rehabilitation has not been activated yet in Korea, so the number of specialists dedicated to the cardiac rehabilitation team is very small. However, to find the purpose of this study, we think that an investigation into the current state would also be meaningful.

Everyone is aware that hospitals with CRP need to expand in the future; hence, a specific roadmap and support strategies should be planned. We believe that this survey’s results will become useful when undertaking these steps. Promoting CRP is not possible with institutional efforts alone, but government policies and financial support also need to be provided. Therefore, more specific and in-depth survey studies should be conducted in the future. 

## 5. Conclusions

In hospitals with CRP, high costs and issues with patient transportation—mode and distance—were identified as the major barriers that inhibit the activation of CRP. Additionally, in hospitals without CRP, all factors were important barriers, especially the items related to equipment, space, workforce and budget. Therefore, in order to actively promote CRP, it is suggested that the government consider the customized support system according to the performance of CRP.

## Figures and Tables

**Figure 1 jcm-11-02560-f001:**
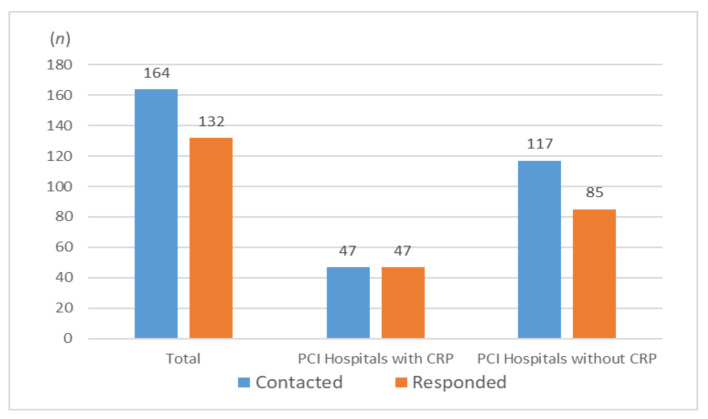
The response rates according to hospitals, with and without cardiac rehabilitation programs (CRP).

**Figure 2 jcm-11-02560-f002:**
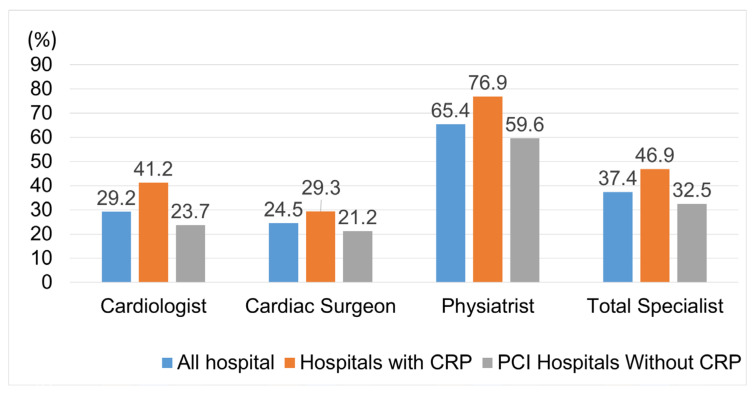
The response rates according to specialties and the presence or absence of cardiac rehabilitation programs (CRP).

**Table 1 jcm-11-02560-t001:** The assessment checklist’s questions for identifying barriers and investigating the implementation status of cardiac rehabilitation.

Question: If Cardiac Rehabilitation Programs (CRP) Are Not Actively Used in Your Respective Hospitals, What Do You Think Would Be the Barriers and Factors for Cardiac Rehabilitation? (Select One for Each Item)					
	Not at All 1 Point	Not Really 2 Points	Neutral 3 Points	Some-What 4 Points	Very Much 5 Points
(1) Lack of Patient Referrals /Lack of Awareness among Specialists					
(2) Lack of Equipment for CRP					
(3) Lack of Space for CRP					
(4) Lack of Workforce for CRP					
(5) Lack of Budget for CRP					
(6) Insufficient Insurance/ High Costs					
(7) Patient Transportation: Mode and Distance					
(8) Additional Opinions					

**Table 2 jcm-11-02560-t002:** The general characteristics between the hospital with and without cardiac rehabilitation programs (CRP).

	Hospitals with CRP (N = 47)	Hospitals without CRP (N = 85)
Performing of Procedure or Surgery (N)		
Percutaneous Coronary Interventions (PCI)	47	85
Coronary Artery Bypass Grafting (CABG)	42	46
Pecutaneous Valve Implantation	42	31
Valvuloplasty	43	42
Endovascular Repair	45	49
Permanent Pacemaker	46	67
Left Ventricular Assis Device (LVAD)	20	11
Annual Number of PCI (N)		
<100	0	11
100~300	6	24
300~500	8	5
500~700	6	10
700~900	8	1
900~1100	6	5
>1100	7	1
unanswered	6	28
Duration of the CRP (N)		
0~1 year	3	85
1~3 year	10	0
3~6 year	15	0
6~9 year	5	0
9~12 year	9	0
>12 year	4	0
unanswered	1	0
No. of Responded Specialist (N)	97	130
Interventional Cardiologist	40	50
Cardiac Surgeon	17	18
Physiatrist	40	62

**Table 3 jcm-11-02560-t003:** Comparison of barrier factors between hospitals with and without cardiac rehabilitation programs (CRP).

	Respondent in Hospitals with CRP (N = 97)	Respondent in Hospitals without CRP (N = 130)	*p* *
Lack of Patient Referrals/Awareness among Physicians	3.11 ± 1.19	3.22 ± 1.09	0.502
Lack of Equipment for CRP	2.14 ± 1.02	4.32 ± 0.78	0.000 *
Lack of Space for CRP	2.33 ± 1.13	4.29 ± 0.81	0.000 *
Lack of Workforce for CRP	2.93 ± 1.16	4.40 ± 0.69	0.000 *
Lack of Budget for CRP	2.87 ± 1.17	4.31 ± 0.72	0.000 *
Insufficient Insurance/High Costs	3.42 ± 0.90	3.80 ± 0.97	0.003 *
Patient Transportation: Mode and Distance	3.77 ± 1.01	3.18 ± 0.98	0.000 *

Values are presented as mean ± standard deviation. *p* * values are for a comparison between hospitals with and without CRP. Statistically significant (*p* < 0.05) differences between the groups.

**Table 4 jcm-11-02560-t004:** Barriers and factors relating to cardiac rehabilitation at hospitals with cardiac rehabilitation programs (CRP).

Hospitals with CRP	Physiatrists (N = 40)	Others Specialists (N = 57)	*p* *
Lack of Patient Referrals/Awareness among Physicians	3.21 ± 1.22	2.97 ± 1.15	0.471
Lack of Equipment for CRP	1.63 ± 0.91	2.47 ± 0.92	0.003 *
Lack of Space for CRP	1.96 ± 1.17	2.50 ± 1.02	0.086
Lack of Workforce for CRP	2.75 ± 1.18	3.23 ± 1.06	0.128
Lack of Budget for CRP	2.54 ± 1.20	3.17 ± 1.03	0.039 *
Insufficient Insurance/High Cost	3.46 ± 0.85	3.37 ± 0.91	0.666
Patient Transportation Mode and Distance	4.25 ± 0.82	3.50 ± 1.02	0.004 *

Values are presented as mean ± standard deviation. *p* * values are for comparison between physiatrists and other specialists (cardiologists and cardiac surgeons). Statistically significant (*p* < 0.05) differences between the groups.

**Table 5 jcm-11-02560-t005:** Barriers and factors relating to cardiac rehabilitation at hospitals without cardiac rehabilitation programs (CRP).

Hospitals without CRP	Physiatrists (N = 62)	Others Specialists (N = 68)	*p* *
Lack of Patient Referrals/Awareness among Physicians	3.26 ± 1.11	3.18 ± 1.05	0.670
Lack of Equipment for CRP	4.48 ± 0.59	4.18 ± 0.89	0.024 *
Lack of Space for CRP	4.48 ± 0.68	4.12 ± 0.85	0.010 *
Lack of Workforce for CRP	4.48 ± 0.67	4.32 ± 0.66	0.186
Lack of Budget for CRP	4.37 ± 0.77	4.25 ± 0.66	0.344
Insufficient Insurance/High Costs	3.73 ± 1.02	3.87 ± 0.91	0.406
Patient Transportation: Mode and Distance	3.10 ± 1.02	3.26 ± 0.91	0.330

Values are presented as mean ± standard deviation. *p* * values are for a comparison between physiatrists and other specialists (cardiologists and cardiac surgeons). Statistically significant (*p* < 0.05) differences between the groups.

## Data Availability

Data supporting this study’s findings are available from the KNIH, but since it was used under license specifically for the current study, restrictions apply, and so, it is not publicly available. Data are however available from the authors upon reasonable request and with KNIH’s permission.

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
