# Peer review of "A Comparison of Barrier Factors between Hospitals with and without Cardiac Rehabilitation Programs in Korea: A Nation-Wide Survey Research"

_jcm, 2022, doi:10.3390/jcm11092560_

Round 1

Reviewer 1 Report

A Comparison of Barrier Factors between Hospitals With and Without Cardiac Rehabilitation Programs in Korea: A nation-wide survey research (jcm-1643359)

Dear authors

Thank you very much for submitting your paper to the Journal of Clinical Medicine. Your aim of the study is to investigate the current status in hospitals that do not provide CRP, and identify the differences in barriers between the hospitals, with or without cardiac rehabilitation programs (CRP). This study is very interesting and get the useful information on clinical practice of referral and healthcare system. A comparison of barrier factors between hospitals with and without cardiac rehabilitation programs in Korea lead to the improvement and campaign of the CRP. I have some comments on your manuscript. 

Major comment

  1. Materials and Methods > 1. Target subjects and Conducting of the Survey
    1. The inclusion and exclusion criteria to select the specialists, interventional cardiologists, cardiac surgeons and physiatrists (such as the experience of specialist) or criterion to select the hospitals should be described in this section.

  1. The general characteristic between the hospital with and without CRP should be described and compared in the Table. (such as experience of PCI, availability of specialist including interventional cardiologists, cardiac surgeons and physiatrists)

  1. Is there the other specialist which should be involved in study? (such as general practice, intervention radiologist, etc.)

  1. The patient’s awareness, cooperation and surveillance in CRP are one of important factor which should be investigated and may important factor and barrier in CRP. Dose the authors was included the patient’s response or discuss about the level of the patient’s cooperation in this manuscript ?

  1. The duty of all specialists in CRP program should be documented in manuscript. Because the different role of each specialists effected to the response and point of questionnaire.

Best wishes

Reviewer  

Author Response

Response to Reviewer #1 Comments

Dear Reviewer #1,

We would like to express our sincere gratitude for the reviewers’ thorough consideration and scrutiny of our manuscript. Due to those informative comments, we noted the areas in which the manuscript could be improved, and we now resubmit a revision of the manuscript and the point-by- point responses to the reviewers' comments.

Major comment

Point 1: Materials and Methods > 1. Target subjects and Conducting of the Survey

The inclusion and exclusion criteria to select the specialists, interventional cardiologists, cardiac surgeons and physiatrists (such as the experience of specialist) or criterion to select the hospitals should be described in this section.

à Response 1: According to Reviewer #1’s recommendation, we have added a describing the detailed criteria to selection of the subjects and target hospital to the methods section as follows;

 (Materials and Methods Section)

Target hospitals were selected when insurance claims of PCI were registered with Health Insurance Review and Assessment Service (HIRA) as an independent agency responsible for the claims review and quality assessment of the National Health Insurance. Specialists have professional licenses recognized by the Minister of Health and Welfare in their field. In particular, interventional cardiologists must have at least two years of experiences and currently performing PCI.

Point 2: The general characteristic between the hospital with and without CRP should be described and compared in the Table. (such as experience of PCI, availability of specialist including interventional cardiologists, cardiac surgeons and physiatrists)

à Response2: According to Reviewer #1’s recommendation, we have added the general characteristic between the hospital with and without CRP to the result section as follows;

(Result Section)

3.1. The general characteristic between the hospital with and without CRP

  • In hospitals with CRP, various procedures and surgeries were performed, and the annual number of PCI is high (Table 2).
  • The period of cardiac rehabilitation in hospitals with CRP was the most between 3 and 6 years (Table 2).

Table 2. The general characteristic between the hospital with and without cardiac rehabilitation programs (CRP).

Hospitals with CRP

(N=47)

Hospitals without CRP

(N=85)

Peforming of Procedure or Surgery (N)

  Percutaneous Coronary Interventions (PCI)

47

85

Coronary Artery Bypass Grafting (CABG)

42

46

Pecutaneous Valve Implantation

42

31

Valvuloplasty

43

42

Endovascular Repair

45

49

Permanent Pacemaker

46

67

Left Ventricular Assis Device (LVAD)

20

11

Annual Number of PCI (N)

< 100

0

11

100~300

6

24

300~500

8

5

500~700

6

10

700~900

8

1

900~1100

6

5

> 1100

7

1

Unanswered

6

28

Duration of the CRP (N)

0 ~ 1 year

3

85

1 ~ 3 year

10

0

3 ~ 6 year

15

0

6 ~ 9 year

5

0

9 ~12 year

9

0

> 12 year

4

0

Unanswered

1

0

No. of Responded Specialist (N)

97

130

Interventional Cardiologist

40

50

Cardiac Surgeon

17

18

Physiatrist

40

62

Point 3: Is there the other specialist which should be involved in study? (such as general practice, intervention radiologist, etc.)

àResponse 3: Currently, general practitioners in Korea have very low interest in cardiac rehabilitation and have very low referral rates, so they are not included. However, as you pointed out, it is regrettable that we did not include various team members such as cardiac rehabilitation exercise therapists. We have added the statement about limitation to the discussion section as follows:

(Discussion Section)

It is also necessary to conduct future research involving various team members such as general practitioners, exercise therapists, nutritionist, etc.

Point 4: The patient’s awareness, cooperation and surveillance in CRP are one of important factor which should be investigated and may important factor and barrier in CRP. Dose the authors was included the patient’s response or discuss about the level of the patient’s cooperation in this manuscript?

à Response 4: I totally agree with the Reviewer #1's comments. I think the patient's response and cooperation are very important parts in cardiac rehabilitation. Another investigation into this government project is included. The results will be summarized and presented in another paper. The investigation that produced this report is focused on investigating the thoughts of specialist working between hospitals with and without CRP, and another investigation is an in-depth analysis of patients using different questionnaire.

Point 5: The duty of all specialists in CRP program should be documented in manuscript. Because the different role of each specialists effected to the response and point of questionnaire.

à Response 5: Thank you for the Reviewer #1’s thoughtful and important comments. In Korea, there are still many places where cardiac rehabilitation has not been activated, so there is a textbook role division, but it is very difficult to accurately state the situation as each hospital is very diverse. However, we added the content to the subject part in the method section and discussion Section.

(Method Section)

The included specialists were not excluded depending on whether they were actually playing a role in the cardiac rehabilitation team.

(Discussion Section)

Another limitation is that the role of the specialist as a cardiac rehabilitation team cannot be presented separately. This is because cardiac rehabilitation has not been activated yet in Korea, so the number of specialists dedicated to the cardiac rehabilitation team is very small. However, to find the purpose of this study, we think that investigation in the current state is also meaningful.

Reviewer 2 Report

This prospective study by Kim et al. investigated the barriers of referring patients for cardiac rehabilitation programs. This is a very interesting and challenging topic, however, extensive modification is needed. The primary concerns regarding this study method of study are not clearly defined. 

  • What was the target group, patients who underwent cardiac cath? If yes, elective or urgent catheterization? What about patients who were hospitalized due to heart failure exacerbation? 
  • How the response rate of hospitals and physicians were counted?
  • Please support this statement in the abstract with data: ‘While the specialists at hospitals with CRP identified patient referral, transportation, and cost as the major barriers, those at hospitals without CRP, did not perceive patient referral, but only transportation and cost as the major barriers‘ Has been the difference statistically significant? Did the authors investigate the answers provided by individual institutions or by individual physicians? What factors were investigated? 
  • How did the authors decide on sending the questionnaire electronically vs. via post? The different forms of requiring information can introduce bias in response and rate and consequently in the main finding of the study.
  • How did the authors define the primary outcome of the study- referral placed for CRP at the discharge or actually starting the rehabilitation program or was it something else? How did the authors measure this outcome eg extracted from electronic medical reports?

Author Response

Response to Reviewer #2 Comments

Dear Reviewer #2,

We would like to express our sincere gratitude for the reviewers’ thorough consideration and scrutiny of our manuscript. Due to those informative comments, we noted the areas in which the manuscript could be improved, and we now resubmit a revision of the manuscript and the point-by- point responses to the reviewers' comments.

General comment

This prospective study by Kim et al. investigated the barriers of referring patients for cardiac rehabilitation programs. This is a very interesting and challenging topic, however, extensive modification is needed. The primary concerns regarding this study method of study are not clearly defined.

à Response: Thank you for the Reviewer #2’s thoughtful and important comments. According to Reviewer #2’s recommendation, we have clearly delineated the method section

Major comment

Point 1: What was the target group, patients who underwent cardiac cath? If yes, elective or urgent catheterization? What about patients who were hospitalized due to heart failure exacerbation?

à Response 1: According to the insightful comments of the Reviewer #2, we have added sentences expressing more clearly and detailed information about the target hospital and specialist to the method section, and have added table about the general characteristics of the hospital to the result section

(Method Section)

Specialists have professional licenses recognized by the Minister of Health and Welfare in their field. In particular, interventional cardiologists must have at least two years of experiences and currently performing PCI.

(Result Section)

3.1. The general characteristic between the hospital with and without CRP

  • In hospitals with CRP, various procedures and surgeries were performed, and the annual number of PCI is high (Table 2).
  • The period of cardiac rehabilitation in hospitals with CRP was the most between 3 and 6 years (Table 2).

Table 2. The general characteristic between the hospital with and without cardiac rehabilitation programs (CRP).

Hospitals with CRP

(N=47)

Hospitals without CRP

(N=85)

Peforming of Procedure or Surgery (N)

  Percutaneous Coronary Interventions (PCI)

47

85

Coronary Artery Bypass Grafting (CABG)

42

46

Pecutaneous Valve Implantation

42

31

Valvuloplasty

43

42

Endovascular Repair

45

49

Permanent Pacemaker

46

67

Left Ventricular Assis Device (LVAD)

20

11

Annual Number of PCI (N)

< 100

0

11

100~300

6

24

300~500

8

5

500~700

6

10

700~900

8

1

900~1100

6

5

> 1100

7

1

Unanswered

6

28

Duration of the CRP (N)

0 ~ 1 year

3

85

1 ~ 3 year

10

0

3 ~ 6 year

15

0

6 ~ 9 year

5

0

9 ~12 year

9

0

> 12 year

4

0

Unanswered

1

0

No. of Responded Specialist (N)

97

130

Interventional Cardiologist

40

50

Cardiac Surgeon

17

18

Physiatrist

40

62

Point 2: How the response rate of hospitals and physicians were counted?

à Response 2: I totally agree with the Reviewer #2's questions and comments. Response rate should be defined. As your comments, we have added the criteria of selected hospitals and criteria for response success.

(Method Section)

Target hospitals were selected when insurance claims of PCI were registered with Health Insurance Review and Assessment Service (HIRA) as an independent agency responsible for the claims review and quality assessment of the National Health Insurance.

If more than 60% of the questions were answered, it was counted as a respondent, and the hospital where the respondent works was also counted as the responded hospital.

Point 3: Please support this statement in the abstract with data: ‘While the specialists at hospitals with CRP identified patient referral, transportation, and cost as the major barriers, those at hospitals without CRP, did not perceive patient referral, but only transportation and cost as the major barriers‘ Has been the difference statistically significant? Did the authors investigate the answers provided by individual institutions or by individual physicians? What factors were investigated?

à Response 3: According to Reviewer #2’s recommendation, we have added and changed the results and their significances to the abstract and conclusions section.

(Abstract Section)

all items are considered as barriers, especially the items related to equipment, space, workforce, and budget are more serious barriers. did not perceive patient referral, but only transportation and cost as the major barriers. Therefore, in order to actively promote CRP, it is suggested that government consider the customized support system according to the performing of the CRP. Therefore, high cost and patient transportation related issues such as mode and distance were identified as the major bar-riers that inhibit the activation of CRP.

(Conclusion Section)

In hospitals with CRP, high costs and issues with patient transportation—mode and distance—were identified as the major barriers that inhibit the activation of CRP. And, in hospitals without CRP, all factors were important barriers especially the items related to equipment, space, workforce, and budget. Therefore, in order to actively promote CRP, it is suggested that government consider the customized support system according to the performing of the CRP. Therefore, the government needs to actively promote CRP by considering insurance fees and improving accessibility to hospitals.

Point 4: How did the authors decide on sending the questionnaire electronically vs. via post? The different forms of requiring information can introduce bias in response and rate and consequently in the main finding of the study.

àResponse 4: Thank you for your insightful comment. In our study, both post and electronic questionnaire were used. This was to increase the response rate in the Corona situation. Most of the responses were sent electronically, a small number by post, and some responded to both post and electronic. We have added an important comment you pointed out in the discussion section.

(Discussion section)

Depending on the response request method, the response rate may be affected and there is a possibility that it may affect the result. However, since the survey was at the time of the outbreak of COVID-19, both post and electronic questionnaire were used to increase the response rate.

Point 5: How did the authors define the primary outcome of the study- referral placed for CRP at the discharge or actually starting the rehabilitation program or was it something else? How did the authors measure this outcome eg extracted from electronic medical reports?

à Response 5: Thank you for your thoughtful question. The results of this study are the specialists' thoughts on the barrier to cardiac rehabilitation, and the question on the questionnaire asked why cardiac rehabilitation is not activated in the hospital to which you belong. This question is focused on the part that does actually perform cardiac rehabilitation, so it seems to be considered as the actually activation and starting of the cardiac rehabilitation program in the hospital. The values answered by the specialist on the questionnaire using a 5-Likert scale were converted into Excel for statistical analysis, and the values were not extracted from the electronic medical record.

Reviewer 3 Report

The authors have done plenty of work to conduct this research and analysis and show overall situation of CRP in the whole Korea. Here are some concerns I couldn't figure out about this study:

1.In line 47, the abstract conveys that, patient referral is the only difference between hospitals with or without CRP, The author didn't give a reasonable inference or explanation for this distinction.

2.For an uninitiated reader, a brief introduction(Including the content of the project, the approximate price and other related information) of CP are needed.

3.In line 85, why the responses are 607 specialists and 164 hospitals but not 224 and 132 as described in the abstract. No response is also a response ?

4.What does Figure 2 want to convey and what does"This finding indicates that an interest in CR was reflected in the response rate. "in line 155 means? Reflected what?

5.In table 2, The number of hospitals with CRP is 97, shouldn't it be 47, and 130 for hospitals without CRP be 85?

6.In table 3 and 4, I couldn't understand the intention to differ the response between physiatrists and other specialists. What the authors want to convey through this difference?

7.In Table 4, transportation is not as important as other factors through all the  involved specialists. It's inconsistent to the abstract and conclusion.

8. For all the specialists involved in Table 3 and 4, the total number is 40+57+62+68=227.Not 224.

9.The difference between urban and local areas should be discussed.

Author Response

Response to Reviewer #3 Comments

Dear Reviewer #3,

We would like to express our sincere gratitude for the reviewers’ thorough consideration and scrutiny of our manuscript. Due to those informative comments, we noted the areas in which the manuscript could be improved, and we now resubmit a revision of the manuscript and the point-by- point responses to the reviewers' comments.

Major comment

The authors have done plenty of work to conduct this research and analysis and show overall situation of CRP in the whole Korea. Here are some concerns I couldn't figure out about this study:

Point 1: In line 47, the abstract conveys that, patient referral is the only difference between hospitals with or without CRP, The author didn't give a reasonable inference or explanation for this distinction.

àResponse 1: As the reviewer #3’s comment, we made a mistake in describing the results in the abstract, so we have changed the description in the abstract section

(Abstract Section)

all items are considered as barriers, especially the items related to equipment, space, workforce, and budget are more serious barriers. did not perceive patient referral, but only transportation and cost as the major barriers.

Point 2: For an uninitiated reader, a brief introduction(Including the content of the project, the approximate price and other related information) of CP are needed.

à Response 2: According to Reviewer #3’s recommendation, we have added a describing the brief introduction of CRP

(Introduction Section)

Cardiac rehabilitation (CR) is a program of exercise, education and counselling designed to help you recover after CVD. The phase II of cardiac rehabilitation program (CRP) last for up to 6-12 weeks, and the price varies greatly in each country. 

Point 3: In line 85, why the responses are 607 specialists and 164 hospitals but not 224 and 132 as described in the abstract. No response is also a response ?

à Response 3: Thank you for your precise comment. We have corrected the sentence containing the wrong number.

(Materials and Method Section)

For statistical analysis, the responses of the 227607 specialists working in 132164 hospitals that performed the PCI were converted into scores,

Point 4: What does Figure 2 want to convey and what does "This finding indicates that an interest in CR was reflected in the response rate. "in line 155 means? Reflected what?

à Thank you for the Reviewer #3’s insightful comments. We have changed the sentence to be more clearly to the discussion section as follows;

(Discussion Section)

This may be the result because the degree of interest and attentiveness in cardiac rehabilitation had an effect on the response rate to increase. This finding indicates that an interest in CR was reflected in the response rate.

Point 5: In table 2, The number of hospitals with CRP is 97, shouldn't it be 47, and 130 for hospitals without CRP be 85?

àResponse 5: Thank you for your thoughtful comments. The number shown in the table 2 is the number of respondents, not the number of hospitals. Therefore, we have clearly changed it so as not to be confused.

(Table 2)

Hospitals with CRP à Respondent in Hospitals with CRP

Hospitals without CRP à Respondent in Hospitals without CRP

Response 6. In table 3 and 4, I couldn't understand the intention to differ the response between physiatrists and other specialists. What the authors want to convey through this difference?

à Response 6: Although there was not much difference between the responding specialists, there were cases where there was a statistical difference in some items. But, the actual score difference was about 0.5, it was thought that it was difficult to give a great clinical significance, so it was only described in the results. However, as it is necessary to interpret the meaning of the result, as your insightful comments, we have added related content to the discussion section

(Discussion section)

In the comparison between specialists, there were several items with statistical differences, but the difference in actual scores was not as large as 0.5, so it is not considered to have great clinical significance. Among the results, in hospitals with CRP, physiatrists answered that the equipment was low as a barrier compared to other specialists, and in hospitals without CRP, the importance of equipment was high.

Point 7: In Table 4, transportation is not as important as other factors through all the involved specialists. It's inconsistent to the abstract and conclusion.

à Response 7: Thank you for pointing out our big mistake. The abstract and conclusion have been modified to be consistent.

(Abstract Section)

all items are considered as barriers, especially the items related to equipment, space, workforce, and budget are more serious barriers. did not perceive patient referral, but only transportation and cost as the major barriers. Therefore, in order to actively promote CRP, it is suggested that government consider the customized support system according to the performing of the CRP. Therefore, high cost and patient transportation related issues such as mode and distance were identified as the major bar-riers that inhibit the activation of CRP.

(Conclusion Section)

In hospitals with CRP, high costs and issues with patient transportation—mode and distance—were identified as the major barriers that inhibit the activation of CRP. And, in hospitals without CRP, all factors were important barriers especially the items related to equipment, space, workforce, and budget. Therefore, in order to actively promote CRP, it is suggested that government consider the customized support system according to the performing of the CRP. Therefore, the government needs to actively promote CRP by considering insurance fees and improving accessibility to hospitals.

  1. For all the specialists involved in Table 3 and 4, the total number is 40+57+62+68=227.Not 224.

à Response 8: Thanks again for your thoughtful and kind comments. As you mentioned, 227 is correct. All erroneous parts of this manuscripts, such as figure 2, abstract and results, have been corrected.

Point 9: The difference between urban and local areas should be discussed.

à Response 9: I totally agree with the Reviewer #3's comments. We think the difference between urban and local area are very important parts in cardiac rehabilitation. But, in Korea, most hospitals performing PCI in the study area located in urban. If there are more hospitals performing PCI in urban, a comparative study will be possible. Therefore, it was difficult to analyze the difference between urban and local. Thank you for the thoughtful advice.

Round 2

Reviewer 3 Report

All my former questions are well addressed. For the language, a English native speaker's revision shall be needed.

Author Response

Response to Reviewer #3 Comments

Dear Reviewer #3,

All my former questions are well addressed. For the language, a English native speaker's revision shall be needed.

Response: We appreciate for your considerate comment. In accordance with your advice, we envelope the certification of English editing by native English speaker.